# Is End-Stage Renal Disease Tumor Suppressive? Dispelling the Myths

**DOI:** 10.3390/cancers16183135

**Published:** 2024-09-12

**Authors:** Toshiro Migita

**Affiliations:** 1Tokyo Nephrology Clinic, Tokyo 170-0003, Japan; t.migita@tokyo-nephro-clinic.com; Tel.: +81-3-3949-5801; 2Division of Cancer Cell Biology, The Institute of Medical Science, The University of Tokyo, Tokyo 108-8639, Japan; 3Department of Medical Laboratory Sciences, Kitasato University, Kanagawa 252-0373, Japan

**Keywords:** end-stage renal disease, carcinogenesis, tumor suppression, cancer immunity, dialysis, uremic immunosuppression

## Abstract

**Simple Summary:**

End-stage renal disease (ESRD) is a life-threatening condition that necessitates renal replacement therapies such as dialysis. ESRD is believed to increase cancer risk and cancer mortality rate; however, data on cancer risk and cancer-specific mortality in patients with ESRD present multiple statistical issues, including sampling errors, bias, and confounding factors, falsely depicting high cancer incidence. Only renal cell and urothelial cancers are driven by ESRD. However, cancer-specific mortality is generally lower in patients with ESRD than in the general population, which corresponds to the fact that cancers arising from ESRD are generally less aggressive and have low metastatic potential. ESRD damages not only normal but also malignant cells in multiple stages of cancer development. This review highlights the potential anticancer effects of ESRD, proposing a reconsideration of the hypothesis that ESRD promotes cancer development and progression.

**Abstract:**

The prevalence of end-stage renal disease is increasing worldwide. Malignancies accompanying end-stage renal disease are detected in approximately 120 individuals per 10,000 person-years. Most studies have suggested that end-stage renal disease causes carcinogenesis and promotes tumor development; however, this theory remains questionable. Contrary to the theory that end-stage renal disease is predominantly carcinogenic, recent findings have suggested that after controlling for biases and sampling errors, the overall cancer risk in patients with end-stage renal disease might be lower than that in the general population, except for renal and urothelial cancer risks. Additionally, mortality rates associated with most cancers are lower in patients with end-stage renal disease than in the general population. Several biological mechanisms have been proposed to explain the anticancer effects of end-stage renal disease, including premature aging and senescence, enhanced cancer immunity, uremic tumoricidal effects, hormonal and metabolic changes, and dialysis therapy-related factors. Despite common beliefs that end-stage renal disease exacerbates cancer risk, emerging evidence suggests potential tumor-suppressive effects. This review highlights the potential anticancer effects of end-stage renal disease, proposing reconsideration of the hypothesis that end-stage renal disease promotes cancer development and progression.

## 1. Introduction

End-stage renal disease (ESRD) is a life-threatening condition. The number of patients with ESRD is increasing worldwide, with a global median incidence of treated ESRD of 146 individuals per million people [1]. Patients with ESRD have a cancer rate of approximately 120 individuals per 10,000 person-years, and the uremic milieu is believed to increase cancer risk [2,3,4]. However, several questions regarding this hypothesis remain unanswered. For instance, although uremia is a systemic pathological state, why does the risk of non-urological cancers remain low in patients with ESRD? Although patients with ESRD undergo frequent examinations to check for heart and lung diseases, why is lung cancer rarely found? Despite cancer being the leading cause of death worldwide in developed countries [5], why are patients with ESRD less likely to die from cancer?

ESRD affects the physiological activities of almost all organs at both cellular and tissue levels. Moreover, it disrupts the homeostasis of the body’s fluid volume, electrolytes, acidity, hormones, energy, and immune system. All these effects influence carcinogenic processes and tumor development. However, no research or review on the potential tumor-suppressive effects of ESRD has been published to date. In this narrative review, I aimed to highlight cancer biology in ESRD and address conflicting issues regarding the dogmatic hypothesis that ESRD causes and promotes cancer. Studies on cancer-specific risk, aggressiveness, and mortality in patients with ESRD were reviewed by carefully selecting reliable recent articles with strict criteria and no biases. Based on the results and latest findings in nephrology and oncology, a potential mechanism of tumor suppression by ESRD is proposed, and the clinical course of cancer in ESRD is estimated.

## 2. Epidemiologic Pitfalls

First, the patient selection criteria for ESRD studies are crucial. To address the precise effect of renal failure on cancer incidence, inclusion criteria should be limited to patients with ESRD undergoing maintenance dialysis therapy. Patients with ESRD diagnosed before or within 6 months of commencing dialysis should be excluded because of extensive examinations and their frequent pre-existing cancers [6].

Second, the analysis of cancer risk in patients with ESRD is subject to various biases, such as selection, confounding, and information biases. In case-control or cohort studies, matching the backgrounds of patients with ESRD and those of the general population is difficult. Patients undergoing regular hemodialysis treatment consult doctors significantly more frequently than the general population does. Moreover, most patients with ESRD are aged > 50 years, which increases their overall cancer risk [7]. Publication bias is also crucial because many early publications reported a very high risk of cancer in patients with ESRD [8,9,10].

Third, patients with ESRD often succumb to cardiovascular and infectious diseases, whereas cancer is a more prevalent cause of death in the general population. In patients with ESRD, de novo cancers develop in those who survive cardiovascular and infectious diseases, thus resulting in a high-value cancer risk. Death should be accounted for as competing with the cancer risk [7].

## 3. Controversies in the Hypothesis

### 3.1. Uremic Immunosuppression

Patients with ESRD are immunocompromised and have a high risk of infection; they also exhibit a reduced response to vaccination. Immunosuppression is believed to contribute to carcinogenesis. However, considerable evidence suggests that infections can counteract cancer by triggering immune responses. In mouse models, various infections induce anticancer effects by non-specific immune stimulation [11]. In humans, the risk of various malignancies is inversely associated with the frequency of fever-inducing infections [12].

Uremic immunosuppression is well known to increase the risk of infections. Epidemiological studies have shown that the risk of bacterial and viral infections in patients undergoing dialysis increases with dialysis vintage; however, the incidence of cancer tends to decline with dialysis vintage [13]. In patients undergoing dialysis, the standardized incidence ratio (SIR) of immune deficiency-related cancers was similar to or lower than that of other cancer types [14]. Although cancer risk in patients with ESRD varies according to the organ site (with the risk being higher in the kidneys and lower in the prostate than that in the general population), it does not correlate with the susceptibility of each organ to bacterial or viral infections. Thus, uremic immunosuppression is unlikely to contribute to carcinogenesis.

### 3.2. Chronic Inflammation

Chronic local inflammation is involved in carcinogenesis; however, the role of chronic systemic inflammation in carcinogenesis remains undetermined. Diabetes mellitus, the current leading cause of ESRD, is believed to increase cancer risk; however, recent reports have raised doubts concerning this opinion [15]. In fact, the risk of cancer in patients with chronic systemic inflammatory diseases such as systemic lupus erythematosus [16], allergic diseases [17], rheumatoid arthritis [18], sarcoidosis [19], familial Mediterranean fever [20], and diabetes mellitus is comparable with or lower than that in healthy controls.

### 3.3. Oxidative DNA Damage

Patients undergoing dialysis experience elevated levels of oxidative DNA damage. Oxidative DNA damage in peripheral lymphocytes is frequently observed in these patients. However, the risk of malignant lymphoma and leukemia in patients with ESRD is lower than that in the general population [21].

Mitochondrial DNA is more susceptible to damage from oxidative stress than nuclear DNA. Mitochondrial mutations are frequently observed in patients with renal cell carcinoma (RCC) with ESRD (ESRD-RCC) [22]. However, no DNA alterations are found in the tumor suppressor genes *FHIT* or *p53* in ESRD-RCC, although both genes are targeted by environmental toxic–mutagenic agents [23].

### 3.4. Accumulation of Carcinogenic Compounds

Accumulation of carcinogenic compounds is observed in patients with ESRD because of impaired renal elimination. Hydroquinone, heterocyclic amines, and N-nitrosodimethylamine are highly suspected carcinogens in patients with ESRD [24,25]. Nevertheless, none of these carcinogens are associated with the predominant cancer sites in patients with ESRD. For example, no significant increase in death from renal cancer was noted in employees involved in the manufacture and use of hydroquinone [26].

### 3.5. Cancer Risk in ESRD

Many early studies with a small number of patients with ESRD have reported a markedly high cancer prevalence or risk compared with cancer risk in patients without ESRD.

For this review, five recent large-scale studies on cancer risk in patients with ESRD from different races and countries were selected to assess cancer risk in ESRD patients (Appendix A) [27,28,29,30,31]. Cancer types are listed according to their frequency in the general population globally [32]. The SIR of overall cancer ranges from 0.94 to 1.6, and that of the top five cancers (lung, breast, colorectum, prostate, and stomach cancers) ranges from 0.27 to 1.65. This supports the notion that a slightly elevated cancer risk in patients undergoing dialysis may occur by chance for breast, lung, prostate, and colon cancers [33]. In contrast, the SIRs of cervical, uterine, thyroid, bladder, and kidney cancers were relatively higher. A high risk of kidney and bladder cancers has also been observed in patients with both chronic renal failure and ESRD in many recent reports [21,27,28,29,30,31]. However, frequent screening for ESRD-mediated secondary hyperparathyroidism may contribute to the overdiagnosis of thyroid cancer. Furthermore, most liver and cervical cancers are considered to be infection-related but not uremia-related. Overall, when considering epidemiological limitations, it is questionable whether ESRD generally causes cancer. Only renal cancers and urothelial cancers (UCs) should be referred to as ESRD-related cancers.

### 3.6. Cancer Aggressiveness and Mortality in ESRD

A stage-matched study showed that the likelihood of a non-localized cancer stage was not significantly different between patients undergoing dialysis and matched controls with lymphomas and cancers of the breast, bladder, colorectum, kidneys, and lungs [34]. The tumor stage and grade of prostate, liver, thyroid, and gastric cancers in patients undergoing dialysis were also comparable with or lower than those in the general population. Importantly, the most common renal cancer is less aggressive in patients with ESRD than in the general population, as described in the following section.

A nationwide survey on mortality in the US and Japan clearly showed a characteristic difference in cancer-related mortality between patients on dialysis and the general population. The cancer mortality rate was reported to be 21.3% in the general US population [35] but 4.1% in patients undergoing dialysis [36]. A similar trend was observed in Japan. Malignancy is currently the leading cause of death in the general Japanese population (mortality rate: 28.5%) [37], and the rate is 9.6% in patients undergoing dialysis [38]. Cancer mortality in the general Japanese population has shown a marked increase, and its rate has almost doubled over the past 30 years (Figure 1a) [37]. In contrast, the increase in cancer mortality among Japanese patients on dialysis was only 3% during this period (Figure 1b) [39]. Reports from other countries have indicated that malignancy is a relatively rare cause of death in patients undergoing dialysis compared with that in the general population [40].

## 4. Renal Cancers and UCs

### 4.1. RCC

The SIR of ESRD-RCC is considerably higher than that of other types of malignancies (Appendix A). Pathologically, ESRD-RCC can be diagnosed as conventional RCC (clear-cell, papillary, and chromophobe RCC) or acquired cystic disease-associated RCC (ACD-RCC) [41]. Genetic profiles suggest that ESRD-RCC mainly consists of two pathological subtypes, clear-cell RCC and papillary RCC, both of which resemble conventional RCC [42]. Therefore, genetic changes alone cannot explain the high risk of ESRD-RCC.

ESRD induces cystic changes in the renal cortex and medulla, resulting in acquired cystic kidney disease (ACKD). ACKD development is associated with extensive tissue damage, severe interstitial inflammation, and post-inflammatory fibrosis. This process may induce error-prone repair and subsequent errors in genetic replication in the epithelium of the cyst. Atypical cysts are the earliest precursors of ACD-RCC. Inherited polycystic kidney disease morphologically resembles ACKD but rarely leads to tumors. The extent and rate of interstitial inflammation and fibrosis around the cysts are more severe in ACD-RCC than in inherited polycystic kidney disease. Thus, the drastic destruction and inflammation of ESRD-mediated tissues may contribute to an increased risk of RCC.

The biological behavior of ESRD-RCC is distinct from that of conventional RCC. Tumor cells in ACD-RCC may proliferate within cysts without invasion, corresponding with the behavior of less aggressive malignant cystic tumors in organs other than the kidneys. ESRD-RCC tumors are smaller, lower grade, and lower stage than conventional RCC tumors [43]. The incidence of metastasis is ≤4.3% in patients with ESRD-RCC [43] and 20–30% in those with conventional RCC [44]. Furthermore, the clinical outcome of ESRD-RCC is more favorable than that of conventional RCC [43].

### 4.2. UC

The SIR of UC is potentially higher than that of other malignancies (Appendix A). The reasons for the high risk of UC in patients with ESRD include chronic irritation, reduced urinary washout effect, and atrophic involution of the bladder. Decreased urine volume and changes in urinary content can induce disuse atrophy of the urinary tract, resulting in local inflammation and subsequent urothelial dysplasia. Moreover, continuous tissue destruction and remodeling may contribute to molecular alterations in inflamed urothelial cells. Thus, UC develops in the inflamed urothelium via a molecular event. Altered gut and urinary microbiomes could also contribute to an elevated incidence of bladder cancer.

One report suggested that the aggressiveness of UC increases with the stage of renal dysfunction [45]. Aggressive pathological variations, including squamous differentiation, are more likely to occur in patients with ESRD than in those without ESRD. Bacterial infection, human papillomavirus infection, and mechanical injury are risk factors for squamous cell carcinoma and squamous cell differentiation. Notably, cancer-specific survival is mostly poor in patients with urinary tract cancer among those with ESRD [46]. This might be related to the limited options for surgical interventions and chemotherapy, as well as the high rate of perioperative complications in patients with ESRD.

## 5. Factors in the Anticancer Effects of ESRD

### 5.1. Premature Aging and Cellular Senescence

Senescence is frequently observed in pre-malignant tumors; therefore, cellular senescence is considered to act primarily as a barrier against carcinogenesis via an intrinsic mechanism [47]. Telomeres and p53 coordinate to regulate carcinogenesis and senescence in human cells. Patients with ESRD show reduced telomerase activity or telomere attrition compared with those in the general population; this tendency is evident in patients undergoing long-term dialysis [48]. Furthermore, the activation of p53 is observed in various types of cells in patients with ESRD. Cancer can develop in patients with ESRD under the same conditions, similar to those observed in centenarians. Cancer incidence, aggressiveness, and mortality reportedly decrease in individuals aged > 80 years [49].

### 5.2. Uremic Fibrosis

Uremic fibrosis is most evident in the kidneys and hearts of patients with ESRD, and it is present in almost all organs. In the absence of cancer, normal tissue fibroblasts and mesenchymal stem cells inhibit cancer initiation, influence epithelial cell differentiation, and limit cancer cell invasion [50]. The uremic milieu induces endoplasmic reticulum stress in epithelial cells and fibroblasts. Endoplasmic reticulum stress subsequently induces endothelial–mesenchymal transition, which represents the loss of epithelial cells and the activation of fibroblasts [51]. Furthermore, reduced numbers of epithelial cells may lower the risk of carcinogenesis.

### 5.3. Altered Cancer Immunity

ESRD modulates various stimulatory and inhibitory factors in the cancer immunity cycle [52] and is expected to have a positive effect on cancer immunity. Patients with ESRD have high levels of proinflammatory cytokines, including interleukin (IL)-6, IL-12, tumor necrosis factor (TNF)-α, and interferon (IFN)-γ, which stimulate cancer antigen presentation [52]. IFN-γ directly kills cancer cells. The levels of β2 microglobulin increase in patients with ESRD, and β2 microglobulin regulates the stability and affinity of an antigen of major histocompatibility class I, facilitating the recognition of cancer cells by T cells.

### 5.4. T Cells

The uremic milieu induces thymic involution; this results in decreased numbers of naïve T cells and increased numbers of proinflammatory memory T cells, contributing to infections and atherosclerotic diseases, respectively. Patients with ESRD have been shown to have a skewed T helper type 1 profile, with a decreased number of T helper type 2 and regulatory T cells [53]. T helper type 1 responses, characterized by T cell production of IFN-γ, IL-2, and TNF-α, are important for tumor rejection. Previous studies have shown that cytotoxic Th cells are increased in patients with ESRD and supercentenarians and possibly exert antitumor effects by recognizing major histocompatibility II in these individuals [54,55].

Many studies have shown that the number of CD8+ T cells, including cytotoxic T lymphocytes, remains unchanged in patients with ESRD compared with that in healthy controls [56]. Programmed cell death protein 1 (PD-1), a major inhibitory receptor of T cell immunity, is highly expressed in T cells in patients with ESRD. In mice with implanted tumor cells, progenitors of exhausted CD8+ T cells, characterized by intermediate PD-1 expression, mediate tumor control [57]. Expression of PD-1 identifies patient-specific antitumor T cell responses in the peripheral blood, and intratumoral PD-1+CD8+ T cells have an intrinsically high capacity for tumor recognition. Thus, the presence of circulating PD-1+CD8+ T cells may provide insights into tumor-resident antitumor lymphocytes.

### 5.5. Natural Killer (NK) Cells

The number of NK cells may decrease in patients undergoing dialysis [58], although there are few studies on NK cells in ESRD cases. Moreover, long-term maintenance of hemodialysis induces the intrinsic activation of NK cell lytic potential [58]. Cancer risk in patients undergoing dialysis decreases with dialysis vintage, suggesting that activated NK cells are involved in this process [13].

### 5.6. Neutrophils

The number of myeloid cells, including neutrophils, is higher than that of lymphoid cells in patients with ESRD. Cancer cells are eliminated by N1-type neutrophils via multiple mechanisms, including the production of oxygen radicals and proinflammatory cytokines and the activation of cytotoxic T lymphocytes [59].

Transforming growth factor (TGF)-β polarizes neutrophils to the protumoral N2 type, whereas TGF-β inhibition recruits a population of antitumoral N1 type. As the circulating TGF-β levels are reportedly lower in patients with ESRD than in healthy controls [60], antitumor N1 neutrophils are expected to be dominant in patients with ESRD.

### 5.7. Monocytes/Macrophages

Circulating monocytes in patients with ESRD are associated with elevated expression of toll-like receptor-2 and toll-like receptor-4 and produce cytokines and reactive oxygen species that contribute to oxidative stress and systemic inflammation [61]. In chronic kidney disease, M1-type macrophages infiltrate the kidneys owing to progressive injury and persistent inflammation. The ratio of M1/M2 macrophages is high in patients on dialysis, and M1 macrophages produce high levels of nitric oxide and type 1 cytokines, both of which are cytotoxic to tumor cells.

### 5.8. Uremic Solutes

In vitro, indoxyl sulfate, a representative protein-bound uremic solute, reduces the viability of HepG2 liver cancer cells and is associated with a decrease in the hypoxia-inducible factor-alpha protein level. In vivo, indoxyl sulfate reportedly had cytostatic effects on breast cancer cells and inhibited cancer invasion and metastasis in a murine model via oxidative stress and activation of the aryl hydrocarbon and pregnane X receptors. 

Uremic solutes induce platelet dysfunction, which increases the risk of hemorrhagic complications in patients undergoing dialysis. Platelets cloak cancer cells and protect them from cancer immunosurveillance [62]. Therefore, ESRD-mediated platelet dysfunction may reduce the metastatic potential of cancer cells.

### 5.9. Changes in Various Hormones

The kidneys regulate sex hormones, and the hypothalamic–pituitary–gonadal axis is affected in ESRD. This is consistent with the evidence of a markedly lower risk and mortality rate of sex hormone-dependent cancers in patients with ESRD [21,31].

Patients undergoing dialysis usually have high levels of atrial natriuretic peptide owing to fluid overload. Both in vitro and in vivo experiments have shown that atrial natriuretic peptide inhibits tumor growth by inhibiting the mitogen-activated protein kinase signaling pathway [63].

The α-Klotho level typically decreases in patients undergoing dialysis and in a wide range of malignancies, suggesting its role as a tumor suppressor. In contrast to the role of α-Klotho, recent studies have shown the protumor effects of β-Klotho and γ-Klotho in several types of cancer [64]. Fibroblast growth factor receptor-4 and β-Klotho or γ-Klotho bind to fibroblast growth factor 19 and activate multiple downstream signaling pathways, including mitogen-activated protein kinase, phosphoinositide-3 kinase, and Janus kinase/signal transducers and activators of transcription.

### 5.10. Periodic and Systemic Acidosis

Malignant transformation is associated with proton transport into the extracellular space, with cancer cells maintaining a high intracellular pH [65]. As increased intracellular and decreased extracellular pH are essential for malignant transformation, an acidic microenvironment may be an obstacle to malignant transformation. Acidotic treatment against cancer was proposed decades ago and has been proven to be effective for various types of cancer in vivo. Exercise-induced transient systemic acidosis delays the progression of an invasive cancer phenotype in breast cancer [66]. Thus, periodic and systemic acidosis in ESRD may be detrimental to cancer development and growth.

### 5.11. Hemodialytic Procedure

Hemodialysis therapy can eliminate cancer cells. The incidence of RCC metastases is lower in patients on hemodialysis than in those not on hemodialysis [67], implying that the dialytic membrane can block or inactivate circulating tumor cells (CTCs). The dialysis membrane breaks apart CTCs, inducing the release of cancer cell antigens into the circulation and stimulating cancer immunity. In clinical settings, hemodialysis has been proposed as a treatment option for eliminating metastatic tumor cells in patients with RCC, melanoma, gastric cancer, and breast cancer [67,68,69,70].

A meta-analysis of the general population revealed that anticoagulation extensively decreased overall mortality in patients with cancer, irrespective of renal function [71]. Clinical trials and experimental studies have been proposed to investigate the anticancer effects of anticoagulants, including antiproliferation, anti-metastasis, drug resistance regulation, and immune modulation.

## 6. Natural History of Cancer in ESRD

ESRD damages normal, pre-malignant, and malignant cells. The primary cellular reactions in normal epithelial cells in patients with ESRD are senescence, growth inhibition, and apoptosis. Even with a normal metabolic status, not all CTCs can lead to metastases, and the uremic milieu could promote the apoptosis of CTCs. Metastatic cancer cells require a niche for metastasis; however, the metastatic niche is impaired because of fibrosis and calcification caused by ESRD. Thus, based on Paget’s seed and soil hypothesis [72], seeds are infirm, and the soil is sterile because of uremia. Moreover, the hematological condition of the circulation (i.e., water) is detrimental to CTCs. Thus, cancer cells do not grow and progress well with impaired seed, soil, or water (Figure 2).

Based on the information above, a model for the clinical course of cancer development in ESRD was proposed by modifying the Welch and Black model (Figure 3) [73]. Malignant transformation occurs frequently, and cancer cells that evade immune surveillance proliferate rapidly under physiological conditions. The number of cancer cells increases over time, spreading to other organs and ultimately resulting in the death of the patient. Malignant transformation can also occur in the uremic milieu. However, its risk is generally assumed to be low except for renal cancers and UCs, which are predominantly caused by drastic physicochemical damage to the renal and urinary systems. Cancer cells in patients with ESRD proliferate slowly and exhibit less invasive and aggressive behavior.

## 7. Conclusions

Previous studies have reported that patients with ESRD have a higher cancer risk and increased cancer-related mortality compared to the general population, which, importantly, led us to believe that ESRD promotes cancer incidence and progression. However, there is no in vitro or in vivo evidence for the pro-cancerous effects of ESRD. A rigorous epidemiological study on patients with ESRD and cancer showed that most ESRD-associated cancers develop independent of uremia, with a relatively low risk. The risk of urological cancers is considerably higher in patients with ESRD than in the general population; however, this is specifically associated with drastic tissue disruption and remodeling of the renal and urinary systems. Cancer metastasis is a relatively rare event in patients with ESRD, and the cancer-related mortality rate is lower than that in the general population. This review provides new insights into the effect of ESRD on cancer biology and prompts reconsideration of the etiology and treatment strategies for cancer in patients with ESRD.

## Figures and Tables

**Figure 1 cancers-16-03135-f001:**
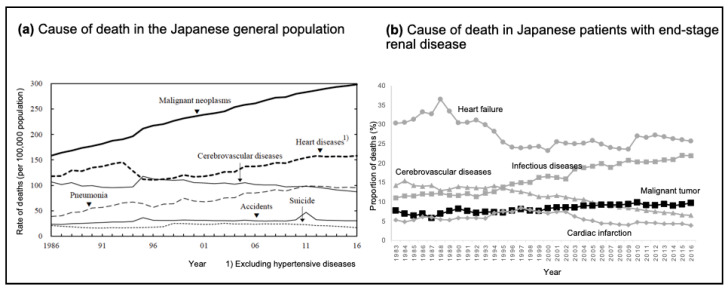
Trends in mortality in the general population and patients with end-stage renal disease in Japan. (**a**) Line graph depicting the rate of deaths per 100,000 people from various causes in the Japanese population between 1986 and 2016. Data are extracted from the Statistical Handbook of Japan 2017 [37]. Reproduced with permission from the Statistics Bureau, Ministry of Internal Affairs and Communications; published by Statistics Japan, 2017. (**b**) Line graph depicting the percentage of deaths in patients on dialysis from various causes in Japan between 1983 and 2016. Data are extracted from the publication of Masakane et al. [38]. Reproduced with permission from Renal Replacement Therapy; published by Springer Nature, 2018.

**Figure 2 cancers-16-03135-f002:**
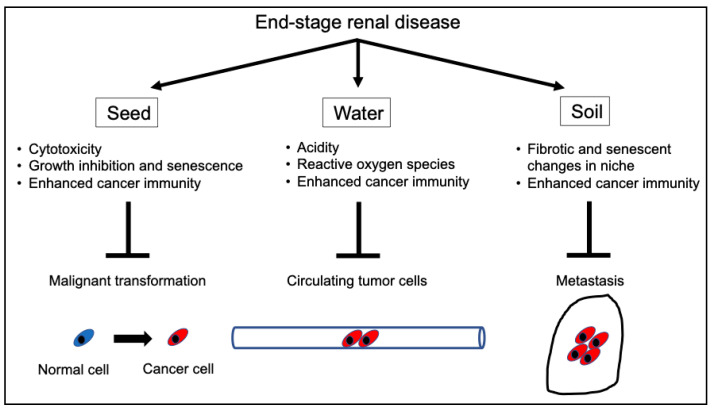
Mechanisms of the tumor-suppressive effects of ESRD. ESRD inhibits cancer initiation and progression at multiple stages. ESRD inhibits carcinogenesis in early cancerous lesions via tumoricidal effects, growth inhibition, and senescence. Moreover, it eliminates circulating cancer cells via cytotoxicity and/or activated cancer immunosurveillance and enhances aging and fibrosis in most organs, preventing cancer cells from metastasizing to distant organs. Overall, ESRD can affect all stages of tumor development. ESRD, end-stage renal disease.

**Figure 3 cancers-16-03135-f003:**
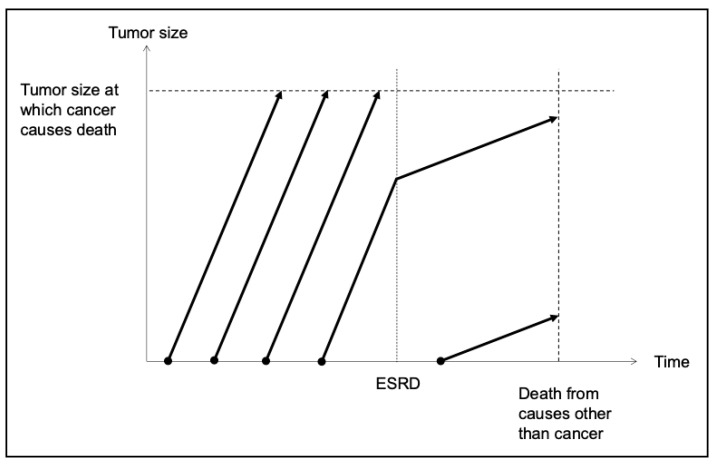
Clinical course of cancer in patients with ESRD. Cancer develops at a relatively high frequency among the healthy population and grows rapidly. These patients ultimately die from cancer with a maximum tumor burden. A certain number of patients with cancer will have ESRD independent of the cancer. In these cases, the growth of cancer cells slows down following ESRD development owing to various mechanisms described in Figure 2. In the presence of ESRD, cancer develops less frequently because carcinogenesis is inhibited at almost all potential sites of cancer development. Nevertheless, once cancer develops in the ESRD milieu, the growth of cancer cells is usually slow, and metastasis to other organs rarely occurs. Although patients with ESRD develop concomitant cancer, most of them die from ESRD-related comorbidities and not from cancer. ESRD, end-stage renal disease.

## Data Availability

No new data were created or analyzed in this study.

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
