# Peer review of "Is End-Stage Renal Disease Tumor Suppressive? Dispelling the Myths"

_cancers, 2024, doi:10.3390/cancers16183135_

Round 1

Reviewer 1 Report

Comments and Suggestions for Authors

Comprehensive review bringing together evidence dispelling the myth that end-stage kidney disease is associated with higher rate of cancer and higher mortality due to cancer and demonstrating the arguments for just the opposite view (with the exception of renal and urothelial cancer). Mechanisms of supposed tumor suppressive effect of uremia is also discussed.

This is really an excellent paper covering all important topics and revolutionizing our view of the relation between end-stage kidney disease and cancer

Comments:

1.       The authors discuss almost exclusively patients on hemodialysis. Could they also comment on the putative difference in cancer risk and outcome between HD and PD?

2.       I would appreciate some comments on the treatment. How the „tumor-suppressive“ effect of end-stage kidney disease be translated into our approach to treatment? Should the treatment of cancer in ESKD be modified (not only avoiding toxicity of overdosing the patients) targeting a bit different role of individual mechanisms of the disease in ESKD?

Author Response

Comment 1. The authors discuss almost exclusively patients on hemodialysis. Could they also comment on the putative difference in cancer risk and outcome between HD and PD?

Response 1. Thank you for your question regarding the differences in cancer risk and outcomes between HD and PD. Several studies have described the better cancer risk and prognosis of PD compared with HD, but they have small samples and are unreliable. The following article is the most reliable and shows no significant difference in outcomes between PD and HD.

https://www.sciencedirect.com/science/article/pii/S0085253815304117

Because no significant difference between HD and PD has been found, and the editor-in-chief has been clear regarding avoiding unnecessary information and references, I have not mentioned it in this paper. I would appreciate the reviewer’s understanding regarding this. 

Comment 2. I would appreciate some comments on the treatment. How the „tumor-suppressive“ effect of end-stage kidney disease be translated into our approach to treatment? Should the treatment of cancer in ESKD be modified (not only avoiding toxicity of overdosing the patients) targeting a bit different role of individual mechanisms of the disease in ESKD?

Response 2. Thank you for your comment. In my experience, clinicians do not actively recommend surgery for cancer in patients with ESKD. However, biologically, because cancers in patients with ESKD metastasize to other organs less often, surgical options are highly recommended. Particularly, nephrectomy is highly recommended for kidney cancer, even in the presence of metastasis. Chemotherapy can be useful for advanced cancers in patients with ESKD, but I would suggest that the growth rate of metastatic tumors is relatively slow, so a watching policy can be applied in such patients.

Reviewer 2 Report

Comments and Suggestions for Authors

Toshiro Migita reviews the impact of ESRD in carcinogenesis. The review is good and provides evidence to have a second thought about the relationship of ESRD and cancer occurrence and if it is due to uremia or cellaular and structural features. No special comment, except two small points: ST1: Numbering should be corrected: Number 10 is followed by 14. 

It would be good to provide a few references for the section of epidemiological pifalls. The claims are strong and some references will help to show that they are not persoanl opinion, or if they are, it should be clarified. 

Author Response

Comment 1. No special comment, except two small points: ST1: Numbering should be corrected: Number 10 is followed by 14. 

Response 1. Thank you for pointing this out to us. I wanted to show the data on the top 10 cancers globally and compared with the data on kidney cancer specifically. Thus, I deleted numbers 11 to 13 in Supplemental Table 1. Because this is easily misunderstood, I added the following: “Data not shown for ranks 11 to 13”.

Comment 2. It would be good to provide a few references for the section of epidemiological pifalls. The claims are strong and some references will help to show that they are not persoanl opinion, or if they are, it should be clarified. 

Response 2. Thank you for this thoughtful suggestion. I have added five references (numbers 6–10) to strengthen and support this section. I also added the following sentence to the end of this section: ‘Death should be accounted for as a competing risk for the cancer risk [7].